# LightGaussian: Unbounded 3D Gaussian Compression with 15x Reduction and 200+ FPS

Zhiwen Fan[1][†][*], Kevin Wang[1][*], Kairun Wen[2], Zehao Zhu[1], Dejia Xu[1], Zhangyang Wang[1]

[1]The University of Texas at Austin    [2]XMU

{zhiwenfan,kevinwang.1839,atlaswang}@utexas.edu

**Project Website**: https://lightgaussian.github.io

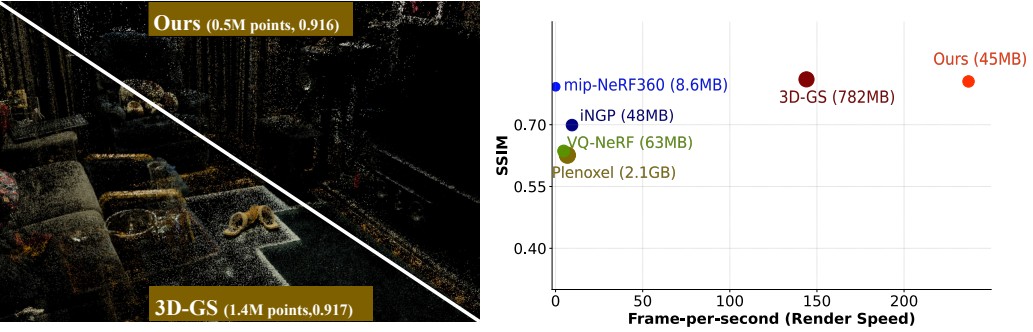

Figure 1: **Compressibility and Speed**: *LightGaussian* compacts 3D Gaussians, reducing storage from 782MB to 45MB and boosting FPS from 144 to 237, while maintaining visual quality.

## Abstract

Recent advances in real-time neural rendering using point-based techniques have enabled broader adoption of 3D representations. However, foundational approaches like 3D Gaussian Splatting impose substantial storage overhead, as Structure-from-Motion (SfM) points can grow to millions, often requiring gigabyte-level disk space for a single unbounded scene. This growth presents scalability challenges and hinders splatting efficiency. To address this, we introduce *LightGaussian*, a method for transforming 3D Gaussians into a more compact format. Inspired by Network Pruning, LightGaussian identifies Gaussians with minimal global significance on scene reconstruction, and applies a pruning and recovery process to reduce redundancy while preserving visual quality. Knowledge distillation and pseudo-view augmentation then transfer spherical harmonic coefficients to a lower degree, yielding compact representations. Gaussian Vector Quantization, based on each Gaussian's global significance, further lowers bitwidth with minimal accuracy loss. LightGaussian achieves an average **15× compression rate** while boosting **FPS from 144 to 237** within the 3D-GS framework, enabling efficient complex scene representation on the Mip-NeRF 360 and Tank & Temple datasets. The proposed Gaussian pruning approach is also adaptable to other 3D representations (e.g., Scaffold-GS), demonstrating strong generalization capabilities.

## 1 Introduction

Novel view synthesis (NVS) aims to generate photo-realistic images of a 3D scene from unobserved viewpoints, given a set of calibrated multi-view images. This capability has widespread applications

---

[*]Z. Fan and K. Wang contributed equally; [†] Z. Fan is the Project Lead

38th Conference on Neural Information Processing Systems (NeurIPS 2024).

in virtual reality [1], augmented reality [2], digital twins [3], and autonomous driving [4]. Neural Radiance Fields (NeRFs) [5–7] have shown promise in 3D modeling from multi-view images by mapping 3D locations and view directions to view-dependent color and volumetric density, with pixel intensity rendered through volume rendering [8]. However, NeRF and its variants face limitations in rendering speed, limiting their deployment in real-world scenarios. To address this, voxel-based representations [9–13], hash grids [14], and neural light fields [15] have been developed. Despite improvements, these methods often compromise between quality and speed.

Recent progress in point-based 3D Gaussian Splatting (3D-GS) [16] has enabled real-time rendering with photo-realistic quality for complex scenes. By representing the scene with explicit 3D Gaussians and using a splatting technique [17], 3D-GS balances speed and quality, making it suitable for large-scale scenarios like digital twins and autonomous driving. However, point-based methods incur high storage costs, as each point and its attributes require independent storage. Additionally, heuristic densification of sparse SfM points into dense Gaussians often results in overparameterization, leading to excessive storage and slower rendering speeds. For instance, a typical unbounded 360-degree scene in 3D-GS [7] may require over 1GB of storage (e.g., 1.4GB for the *Bicycle* scene).

In this paper, we address **storage and rendering speed** issues by developing a compact representation that retains the original rendering quality. The heuristic densification process in 3D-GS results in significant redundancy. Our method, *LightGaussian*, reduces redundancy by targeting both Gaussian count (N) and feature dimension (F) through a comprehensive pipeline:

- To reduce Gaussian count (N), we propose a *Gaussian Pruning and Recovery* step, identifying and removing Gaussians with minimal impact on visual quality, followed by recovery to ensure smooth adaptation.

- For compressing features (F), we introduce an *SH Distillation* process to compact higher-degree spherical harmonic (SH) coefficients, supported by pseudo-view augmentation. Additionally, we employ *Vector Quantization* (VQ) to adaptively select a codebook of Gaussian attributes (e.g, positions, scales, and rotations), reducing precision for less significant features and applying quantization-aware fine-tuning to maintain quality.

In summary, **LightGaussian** achieves substantial compression (e.g., from 782MB to 45MB) while maintaining visual fidelity (SSIM decrease of only 0.007 on Mip-NeRF 360). Rendering speed also improves, reaching over 200 FPS on complex scenes. Our Gaussian Pruning and Recovery method generalizes well, enhancing performance across different 3D Gaussian formats, such as Scaffold-GS.

## 2    Related Works

**Efficient 3D Scene Representations.**    Neural radiance fields (NeRF) [5] use multi-layer perceptrons (MLPs) to represent scenes, setting new standards for view synthesis quality. However, NeRF models face challenges with slow inference speeds, limiting practical use. Efforts to address this have explored ray re-parameterizations [6, 7], explicit spatial data structures [18–21, 9, 14, 13], caching and distillation techniques [15, 22–24], and ray-based representations [25, 26]. Nevertheless, achieving real-time rendering remains challenging for NeRF methods, especially for large-scale scenes where multiple queries per pixel hinder performance.

Point-based representations, like Gaussians, have been explored in various applications, such as shape reconstruction [27], molecular modeling [28], and point-cloud replacement [29], as well as in shadow [30] and cloud rendering [31]. Pulsar [32] demonstrated efficient sphere rasterization, while 3D Gaussian Splatting (3D-GS) [16] applies anisotropic Gaussians [33] with tile-based sorting to achieve real-time speed and quality comparable to MLP-based methods like Mip-NeRF 360 [7].

Despite its strengths, 3D-GS has high storage demands due to the extensive attributes stored with each Gaussian, often requiring gigabytes per scene and hindering rendering efficiency. Recent concurrent works address these issues by using region-based vector quantization [34], K-means codebooks [35], view-direction exclusion [36], or learned binary masks and grid-based neural networks instead of spherical harmonics (SHs) [37] to reduce model size.

**Pruning and Quantization.**    Model pruning reduces neural network complexity by removing non-significant parameters, balancing performance and resource use. Unstructured [38] and structured

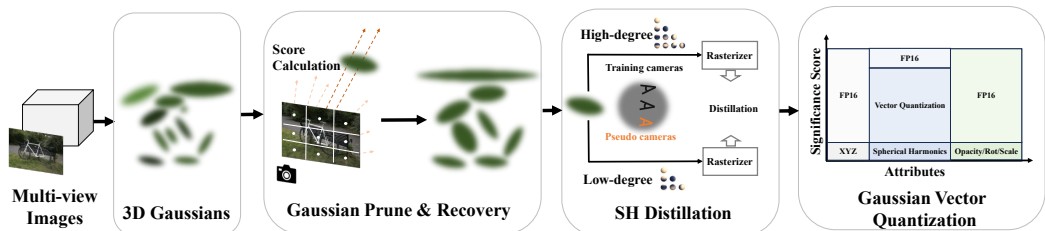

Figure 2: **Pipeline of LightGaussian.** 3D Gaussians are optimized from multi-view images and SfM points. LightGaussian calculates each Gaussian's global significance on training data, pruning those with the least significance. Next, SH coefficients are distilled into a compact format using synthesized pseudo-views. Finally, vector quantization, including codebook initialization and assignment, reduces model bandwidth.

pruning [39, 40] eliminate parameters at the weight level and neuron/channel levels, resulting in a smaller, more efficient architecture. Iterative magnitude pruning (IMP), where low-magnitude weights are progressively removed, has proven effective in methods like lottery ticket rewinding [41, 42].

Vector quantization (VQ) [43] compresses data by representing it with discrete codebook entries (tokens), using mean square error (MSE) to select the closest match in the codebook for each data vector. Prior studies [44–46] have shown that learning discrete, compact representations enhances visual understanding and model robustness. VQ has thus been widely applied in image synthesis [47], text-to-image generation [48], and novel view synthesis [18, 49, 50].

**Knowledge Distillation.** Knowledge Distillation (KD) [51–54, 15] trains a smaller student model by transferring knowledge from a larger teacher model [55]. In 3D vision, KD has been applied to neural scene representations, leveraging view renderings to incorporate 2D priors. For example, DreamFusion [56] and NeuralLift-360 [57] use pre-trained diffusion models for 3D generation, while models like DFF [58], NeRF-SOS [59], INS [60], and SA3D [61] distill 2D image feature extractors to 3D tasks. KD has also been central to compressing scene representation models [15, 24].

## 3 Methods

**Overview.** The LightGaussian framework is illustrated in Fig. 2. The 3D-GS model is trained on multi-view images, initialized from SfM point clouds, and expanded to millions of Gaussians to represent the scene comprehensively. Our pipeline then processes the 3D-GS model into a compact format using *Gaussian Prune and Recovery* to reduce the number of Gaussians, *SH Distillation* to eliminate redundant SHs while retaining key lighting information, and *Vector Quantization* to store Gaussians with lower bit-width.

### 3.1 Background: 3D Gaussian Splatting

3D Gaussian Splatting (3D-GS) [16] is an explicit point-based 3D scene representation, utilizing Gaussians with various attributes to model the scene. When representing a complex real-world scene, 3D-GS is initialized from a sparse point cloud generated by SfM, and *Gaussian Densification* is applied to increase the Gaussian counts that are used for handling small-scale geometry insufficiently covered and over reconstruction. Formally, each Gaussian is characterized by a covariance matrix $\mathbf{\Sigma}$ and a center point $\mathbf{X}$, which is referred to as the mean value of the Gaussian:

$$G(\mathbf{X}) = e^{-\frac{1}{2}\mathbf{X}^T\mathbf{\Sigma}^{-1}\mathbf{X}}, \mathbf{\Sigma} = \mathbf{RSS}^T\mathbf{R}^T, \tag{1}$$

where $\mathbf{\Sigma}$ can be decomposed into a scaling matrix $\mathbf{S}$ and a rotation matrix $\mathbf{R}$.

The complex directional appearance is modeled by an additional property, Spherical Harmonics (SH), with $n$ coefficients, $\{c_i \in \mathbb{R}^3 | i = 1, 2, \ldots, n\}$ where $n = D^2$ represents the number of coefficients of SH with degree $D$. A higher degree $D$ equips 3D-GS with a better capacity to model the view-dependent effect but causes a significantly heavier attribute load.

When rendering 2D images from the 3D Gaussians, the technique of splatting [62, 17] is employed for the Gaussians within the camera planes. With a viewing transform denoted as $\mathbf{W}$ and the Jacobian

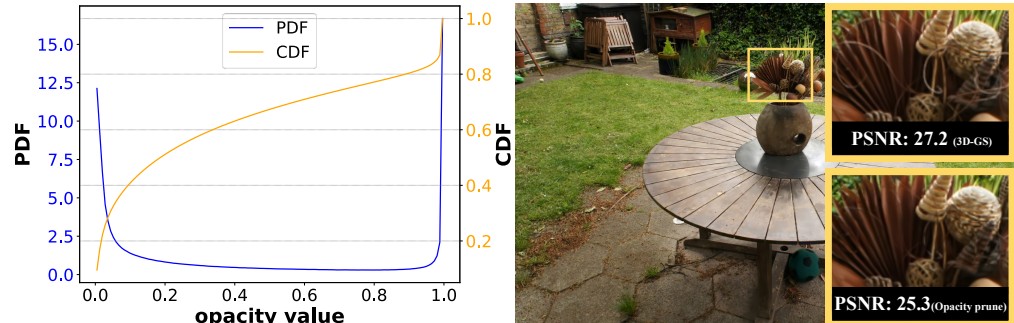

Figure 3: **Zero-shot Opacity-based Pruning**. A significant number of Gaussians exhibit small opacity values (top). Simply utilizing Gaussian opacity as an indicator for pruning the least important Gaussians results in the rendered image losing intricate details (bottom), with the PSNR dropping from 27.2 to 25.3. This has inspired us to find better criteria to measure global significance in terms of rendering quality. The accumulated Probability Density Function(PDF) is equal to 1.

of the affine approximation of the projective transformation represented by $\boldsymbol{J}$, the covariance matrix $\boldsymbol{\Sigma}'$ in camera coordinates can be computed as $\boldsymbol{\Sigma}' = \boldsymbol{JW\Sigma W}^T\boldsymbol{J}^T$. Specifically, for each pixel, the color and opacity of all the Gaussians are computed using the Gaussian's representation Eq. 1. The blending of $N$ ordered points that overlap the pixel is given by the formula:

$$\boldsymbol{C} = \sum_{i \in N} c_i \alpha_i \prod_{j=1}^{i-1} (1 - \alpha_i). \tag{2}$$

Here, $c_i$, $\alpha_i$ represents the view-dependent color and opacity, calculated from a 2D Gaussian with covariance $\boldsymbol{\Sigma}$ multiplied by an optimizable per-3D Gaussian's opacity. In summary, each Gaussian point is characterized by attributes including: position $\boldsymbol{X} \in \mathbb{R}^3$, color defined by spherical harmonics coefficients $\mathrm{SH} \in \mathbb{R}^{(k+1)^2} \times 3$ (where $k$ represents the degrees of freedom), opacity $\alpha \in \mathbb{R}$, rotation factor $\mathbf{R} \in \mathbb{R}^4$, and scaling factor $\mathbf{S} \in \mathbb{R}^3$.

### 3.2 Gaussian Pruning & Recovery

Gaussian densification [16], which clones and refines the initial SfM point cloud, enhances small-scale geometry and detailed scene appearance by improving coverage. While this approach significantly boosts reconstruction quality, it increases the number of Gaussians from thousands to *millions* after optimization, resulting in substantial storage demands.Inspired by neural network pruning techniques [63] that remove less impactful neurons while preserving overall performance, we propose a tailored pruning strategy for 3D Gaussian Splatting to reduce over-parameterized points while maintaining accuracy. Identifying redundant yet recoverable Gaussians is essential to our approach. However, pruning Gaussians based on simple criteria (e.g., point opacity) risks degrading modeling performance, as it may eliminate essential scene details, as shown in Fig. 3.

**Global Significance Calculation.** Inspired by Eq. 2, we go beyond using Gaussian opacity alone to assess significance by evaluating 3D Gaussians within the view frustum, projecting them onto the camera viewpoint for rendering. The initial significance score of each Gaussian (denoted as $\mathrm{G}(\boldsymbol{X}_j)$) can then be quantified based on its contribution to each pixel (ray, $r_i$) across all training views using the criteria $\mathbb{1}(\mathrm{G}(\boldsymbol{X}_j), r_i)$ whether they intersect or not. Consequently, we iterate over all training pixels to calculate the hit count of each Gaussian. The score is further refined by the adjusted 3D Gaussian's volume $\gamma(\boldsymbol{\Sigma}_j)$ and 3D Gaussian's opacity $\sigma_j$, as they all contribute for the rendering formula. The volume calculation equation of the $j$-th 3D Guassian is $\mathrm{V}(\boldsymbol{\Sigma}_j) = \frac{4}{3}\pi abc$, where $abc$ are the 3 dimensions of Scale ($\mathbf{S}$). We also consider the transmittance $\mathbf{T}$ is calculated with one subtract all the 3D Gaussian's opacity the ray hit before the $j$-th 3D Gaussian, $\mathbf{T} = \prod_{i=1}^{j-1} (1- \sigma_i)$. Finally, the global significance score can be summarized as:

$$\mathrm{GS_j} = \sum_{i=1}^{MHW} \mathbb{1}(\mathrm{G}(\boldsymbol{X}_j), r_i) \cdot \sigma_j \cdot \mathbf{T} \cdot \gamma(\boldsymbol{\Sigma}_j), \tag{3}$$

where $j$ is the Gaussian index, $i$ denotes a pixel, and $M$, $H$, and $W$ represent the number of training views, image height, and width, respectively. $\mathbb{1}$ is an indicator function that determines whether a Gaussian intersects a given ray. However, using Gaussian volume alone tends to overemphasize background Gaussians, leading to excessive pruning of Gaussians modeling fine geometry. Thus, we propose a more adaptive approach to measuring volume dimensions:

$$\gamma(\mathbf{\Sigma}) = (V_{\text{norm}})^{\beta}, \tag{4}$$

$$V_{\text{norm}} = \min\left(\frac{V(\mathbf{\Sigma})}{V_{\text{max90}}}, 1\right).$$

Here, the calculated Gaussian volume is firstly normalized by the 90% largest of all sorted Gaussians, clipping the range between 0 and 1, to avoid excessive floating Gaussians derived from vanilla 3D-GS. The $\beta$ is introduced to provide additional flexibility.

**Gaussian Co-adaptation.** We rank all Gaussians by their global significance scores to quantitatively guide pruning of the lower-ranked Gaussians. The remaining Gaussians are then jointly adapted by fine-tuning their attributes—without additional densification—to offset the minor loss from pruning. This adaptation is performed using photometric loss on the original training views, aligned with 3D-GS training for 5,000 iterations.

## 3.3 Distilling into Compact SHs

In the uncompressed Gaussian Splat representation, a substantial 81.3 percent of the feature dimension is occupied by Spherical Harmonics (SH) coefficients, requiring (45+3) floating-point values per splat. Directly reducing the SH degree can save disk space but also diminishes surface "shininess" and affects specular reflections.

To balance model size with scene quality, we introduce a knowledge distillation approach. Knowledge is distilled from a teacher model with full-degree SHs to a student model with truncated, lower-degree SHs. Supervision is based on the difference in predicted pixel intensities between the two models, with images synthesized at camera positions by sampling around each training view according to a Gaussian distribution:

$$\mathcal{L}_{\text{distill}} = \frac{1}{HW} \sum_{i=1}^{HW} \|\boldsymbol{C}_{\text{teacher}}(r_i; [\mathbf{R}|\mathbf{t}]) - \boldsymbol{C}_{\text{student}}(r_i; [\mathbf{R}|\mathbf{t}])\|_2^2. \tag{5}$$

$$\mathbf{t}_{\text{pseudo}} = \mathbf{t}_{\text{train}} + \mathcal{N}(0, \sigma^2), \tag{6}$$

where $\mathbf{R}$ and $\mathbf{t}$ denote rendering camera rotation and position, $\mathbf{t}_{\text{pseudo}}$ and $\mathbf{t}_{\text{train}}$ represent the newly synthesized and training camera positions, respectively. $\mathcal{N}$ denotes a Gaussian distribution with mean 0 and variance $\sigma^2$, which is added to the original position to generate the new position.

## 3.4 Gaussian Attributes Vector Quantization

Vector quantization (VQ) clusters voxels into compact codebooks, enabling high compression rates. However, quantizing Gaussian attributes in a point-based, inherently non-Euclidean representation poses notable challenges. Our empirical findings indicate that applying quantization across all elements—especially for attributes like position, rotation, and scale—results in significant accuracy losses and a marked reduction in precision when represented discretely.

We propose applying VQ to the Spherical Harmonics (SH) coefficients, based on the assumption that a subgroup of 3d Gaussians typically exhibits a similar appearance. Fundamentally, VQ segments the Gaussians $\mathcal{G} = \{\mathbf{g}_1, \mathbf{g}_2, \ldots, \mathbf{g}_N\}$ (here we apply it on SH) to the K codes in the codebook $\mathcal{C} = \{\mathbf{c}_1, \mathbf{c}_2, \ldots, \mathbf{c}_K\}$, where each $\mathbf{g}_j, \mathbf{c}_k \in \mathbb{R}^d$ and K $\ll$ N. $d$ means SH dimension. We aim to strike a balance between rendering quality loss and compression rate by leveraging the pre-computed significance score from Eq. 3. Based on this, we apply VQ selectively on the least significant elements in the Spherical Harmonics (SHs). Specifically, we initialize $\mathcal{C}$ via K-means, iteratively sample a batch of $\mathcal{G}$, associates them to the closest codes by euclidean distance, and update each $\mathbf{c}_k$ via moving average rule: $\mathbf{c}_k = \lambda_d \cdot \mathbf{c}_k + (1 - \lambda_d) \cdot 1/\mathbf{T}_k \cdot \sum_{\mathbf{g}_j \in \mathcal{R}(\mathbf{c}_k)} \mathbf{GS}_j \cdot \mathbf{g}_j$, where $\mathbf{T}_k = \sum_{\mathbf{g}_j \in \mathcal{R}(\mathbf{c}_k)} \mathbf{GS}_j$ is the significance score (Eq. 3) which is assigned to the code vector $\mathbf{c}_k$, $\mathcal{R}(\mathbf{c}_k)$ is the set of Gaussians associated to the $k$-th code. $\lambda_d = 0.8$ represents the decay value, which is utilized to update the

| Methods | Mip-NeRF 360 Datasets | | | | | Tank and Temple Datasets | | | | |
|---|---|---|---|---|---|---|---|---|---|---|
| | FPS↑ | Size↓ | PSNR↑ | SSIM↑ | LPIPS↓ | FPS↑ | Size↓ | PSNR↑ | SSIM↑ | LPIPS↓ |
| Plenoxels [12] | 6.79 | 2.1 GB | 23.08 | 0.626 | 0.463 | 11.2 | 2.3 GB | 21.08 | 0.719 | 0.379 |
| INGP-Big [14] | 9.43 | 48MB | 25.59 | 0.699 | 0.331 | 14.4 | 48MB | 21.92 | 0.745 | 0.305 |
| Mip-NeRF 360 [7] | 0.06 | 8.6MB | 27.69 | 0.792 | 0.237 | 0.14 | 8.6MB | 22.22 | 0.759 | 0.257 |
| VQ-DVGO [49] | 4.65 | 63MB | 24.23 | 0.636 | 0.393 | - | - | - | - | - |
| Compressed 3D-GS* [34] | 152 | 28MB | 27.03 | 0.802 | 0.238 | 202 | 17MB | 23.54 | 0.838 | 0.189 |
| Compact 3D-GS [37] | 128 | 48MB | 27.08 | 0.798 | 0.247 | 185 | 39MB | 23.32 | 0.831 | 0.201 |
| 3D-GS [16] | 134 | 734MB | 27.21 | 0.815 | 0.214 | 154 | 411MB | 23.14 | 0.841 | 0.183 |
| 3D-GS* [16] | 144 | 782MB | 27.40 | 0.813 | 0.217 | 106 | 433MB | 23.66 | 0.845 | 0.178 |
| Ours | 237 | 45MB | 27.13 | 0.806 | 0.237 | 357 | 25MB | 23.44 | 0.832 | 0.202 |

Table 1: **Quantitative Comparisons in Real-world Large-scale Scenes:** Voxel-based methods [12, 14] exhibit insufficient capacity for representing large-scale scenes and are unable to achieve real-time performance. Mip-NeRF360 [7] produces the highest visual quality, but requires over 16 seconds to render a single image. Our method strikes a balance among FPS, model size, and rendering quality, achieving the best balance among efficient representations. For fair metric and visual comparison, we re-trained 3D-GS and Compressed 3D-GS [34] using their original code and configurations on our platform (NVIDIA A6000 GPU), with results marked by *.

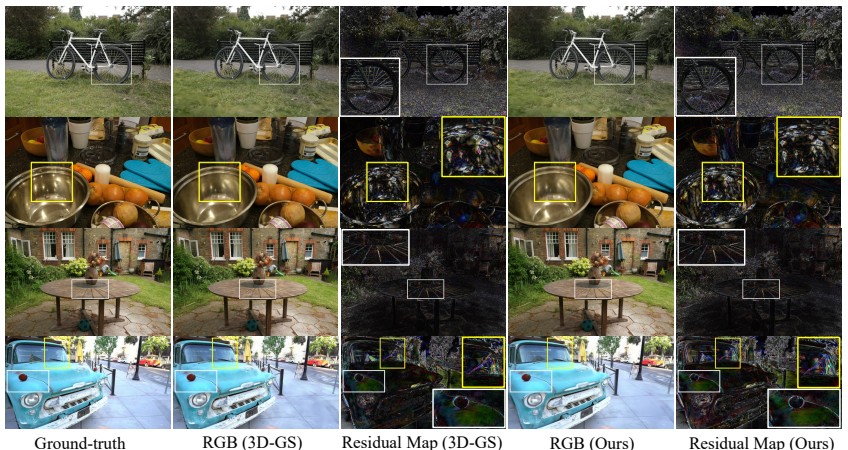

| Ground-truth | RGB (3D-GS) | Residual Map (3D-GS) | RGB (Ours) | Residual Map (Ours) |

Figure 4: **Visual Comparisons**. We compare LightGaussian with the vanilla 3D-GS [16], displaying a residual map between predictions and ground truth scaled from 0 to 127 to emphasize differences. LightGaussian retains most specular reflections (yellow boxes) in its compact format, with a slight change in lightness visible in the bottom white box. The residual maps illustrate discrepancies between rendered images and ground-truth RGB values, where darker areas indicate closer alignment. For a full dynamic viewpoint comparison, please see our supplementary video.

code vector using a moving average. We fine-tune the codebook for 5,000 iterations, while fixing the gaussian-to-codebook mapping. We disable additional clone/split operations and leverage photometric loss on the training views. To preserve essential attributes, including Spherical Harmonics with higher global significance scores, along with Gaussian position, shape, rotation, and opacity, we skip VQ for these elements and store them directly in float16 format.

## 4 Experiments

### 4.1 Experimental Settings

**Datasets and Metrics.** We conduct comparisons using the scene-scale view synthesis dataset provided by Mip-NeRF360 [64], which comprises nine real-world large-scale scenes, including five unbounded outdoor and four indoor settings with complex backgrounds. In addition, we utilize the Tanks and Temples dataset [65], a comprehensive unbounded dataset, and select the same scenes as used in [16]. Performance metrics on synthetic object-level datasets will be detailed in the supplementary materials. We report metrics including the peak signal-to-noise ratio (PSNR), structural similarity (SSIM), and perceptual similarity as measured by LPIPS [66].

**Compared Baselines.** We compare our approach with methods suited for large-scale scene modeling, including Plenoxel [12], Mip-NeRF360 [7], and 3D-GS [16]. Additionally, we evaluate

| Exp# | Model | FPS↑ | Size↓ | PSNR↑ | SSIM↑ | LPIPS↓ |
|---|---|---|---|---|---|---|
| [1] | Baseline (3D-GS [16]) | 156.21 | 365MB | 31.34 | 0.917 | 0.221 |
| [2] | + Gaussian Pruning | 308.10 | 123MB | 30.67 | 0.910 | 0.234 |
| [3] | + Co-adaptation | 304.64 | 123MB | 31.64 | 0.918 | 0.228 |
| [4] | + SH Compactness | 311.58 | 81MB | 30.32 | 0.904 | 0.238 |
| [5] | + photometric loss | 302.20 | 81MB | 31.42 | 0.916 | 0.234 |
| [6] | + Distillation + Pseudo-views | 302.89 | 81MB | 31.48 | 0.917 | 0.231 |
| [7] | + Codebook Quant. | 301.21 | 21MB | 31.09 | 0.918 | 0.236 |
| [8] | + VQ finetune | 302.01 | 21MB | 31.40 | 0.916 | 0.232 |
| [9] | LightGaussian (Ours) | 302.01 | 21MB | 31.40 | 0.916 | 0.232 |

Table 2: Ablation studies on the *Gaussian Pruning & Recovery*, *SH Compactness*, and the *Vector Quantization*. Scene: **Room**. Zero-shot Gaussian pruning leads degraded rendering quality (#2), but Co-adaptation can recover most of the scene details (#3). Directly eliminating high-order SH negatively affects the quality (#4), while distillation with pseudo-view helps to mitigate the gap (#5, #6). Codebook quantization further reduces the required model size and bandwidth (#7, #8).

| Model | FPS↑ | Size↓ | PSNR↑ | SSIM↑ | LPIPS↓ |
|---|---|---|---|---|---|
| Baseline | 156.21 | 365MB | 31.34 | 0.917 | 0.221 |
| Hit Count Only | 301.52 | 123MB | 28.16 | 0.886 | 0.261 |
| + Co-adaptation | 303.43 | 123MB | 30.13 | 0.912 | 0.238 |
| × Opacity | 310.29 | 123MB | 30.27 | 0.909 | 0.239 |
| + Co-adaptation | 304.84 | 123MB | 31.60 | 0.916 | 0.231 |
| × Opacity × $\gamma$(Volume). | 312.30 | 123MB | 30.67 | 0.910 | 0.234 |
| + Co-adaptation | 304.64 | 123MB | 31.64 | 0.918 | 0.228 |

Table 3: Ablation study of the *Gaussian Pruning & Recovery*, by using different Gaussian attributes for computing its global significance score. By considering only the hit count of each Gaussian from training rays, the zero-shot pruning leads to inferior performance. Incorporating the opacity and volume drives us to a better criterion. The subsequent Gaussian Co-adaptation is used to recover most of the information loss from the pruning of redundant Gaussians.

| Model | Size↓ | PSNR↑ | SSIM↑ | LPIPS↓ |
|---|---|---|---|---|
| Baseline | 80.99MB | 31.48 | 0.917 | 0.231 |
| +FP16 | 36.51MB | 31.35 | 0.914 | 0.232 |
| + VQ All att. | 18.74MB | 23.11 | 0.731 | 0.378 |
| + VQ All att. × GS | 19.74MB | 26.23 | 0.826 | 0.323 |
| + VQ SH. | 21.10MB | 30.68 | 0.907 | 0.244 |
| + VQ SH × GS | 21.10MB | 31.16 | 0.911 | 0.235 |
| LightGaussian (w/ VQ Finetune) | 21.10MB | 31.40 | 0.916 | 0.232 |

Table 4: **Ablation Study: *Gaussian Attribute Vector Quantization* (VQ).** Quantizing all attributes to FP16 achieves a smaller model, but applying VQ to all attributes degrades modeling accuracy. Using Global Significance (GS) to target less crucial Gaussians mitigates this loss. Some attributes (e.g., scale) are sensitive to VQ, so we limit VQ to the SH coefficients. By combining VQ with significance scores, LightGaussian achieves an effective balance between model size and quality.

against efficient techniques like Instant-NGP [14], which uses a hash grid for storage efficiency, and VQ-DVGO [49], which extends DVGO [9] with voxel pruning and VQ for optimized representation.

**Implementation Details.** Our framework is implemented in PyTorch and integrates the differentiable Gaussian rasterization technique from 3D-GS [16]. All performance evaluations are conducted on an A6000 GPU. In the Global Significance Calculation phase, we assign a power value of 0.1 in Eq. 4 and proceed to fine-tune the model for 5,000 steps during the Gaussian Co-adaptation process. For SH distillation, we downscale the 3-degree SHs to 2-degree, thereby reducing 21 elements for each Gaussian. This is further optimized by setting $\sigma$ to 0.1 in the pseudo view synthesis stage. In the Gaussian VQ step, the codebook size is configured to 8192, selecting SHs with the least 60% significance score for the vector quantization (VQ ratio), to balance the trade-off between compression efficiency and fidelity.

| | Size ↓ | FPS | PSNR ↑ | SSIM ↑ | LPIPS ↓ |
|---|---|---|---|---|---|
| Scaffold-GS | 173.60 | 152 | 27.96 | 0.8240 | 0.2075 |
| Scaffold-GS + LightGaussian | 112.56 | 178 | 27.78 | 0.8187 | 0.2197 |

Table 5: LightGaussian removes redundant neural Gaussians in Scaffold-GS, reducing model size and improving rendering speed. All experiments were rerun on our platform for fair comparison, with results averaged on the MipNeRF-360 dataset.

## 4.2 Experimental Results

**Quantitative Results.** We assess the performance of various methods for novel view synthesis, with quantitative metrics summarized in Tab. 1. This includes efficient voxel-based NeRFs like Plenoxel [12] and Instant-NGP [14], the compact MLP-based Mip-NeRF360 [7], vector-quantized NeRF [49], and 3D Gaussian Splatting [16].

On the Mip-NeRF360 dataset, MLP-based NeRF methods achieve competitive accuracy with compact representations but suffer from slow inference speeds (0.06 FPS), limiting their practicality. Voxel-based NeRFs improve rendering efficiency but still fall short of real-time performance; for example, Plenoxel requires 2.1GB for a single large-scale scene. In contrast, 3D-GS offers a good balance between quality and speed but demands substantial storage per scene.

Our method, LightGaussian, exceeds existing techniques with rendering speeds over 200 FPS, enabled by efficient rasterization that prunes insignificant Gaussians. This approach reduces 3D Gaussian model redundancy, cutting storage from 782MB to 45MB on Mip-NeRF360—a 15× reduction. LightGaussian also outperforms 3D-GS on the Tank & Temple datasets, nearly doubling rendering speed and reducing storage from 380MB to 22MB.

**Qualitative Results.** We conducted a comparative analysis of rendering results between 3D-GS and LightGaussian, focusing on intricate details and background regions, as shown in Fig. 4. Both 3D-GS and LightGaussian exhibit comparable visual quality, even in challenging scenes with thin structures, showing that LightGaussian effectively removes redundancy while preserving reconstruction fidelity.

**Generalization to Other Point-based Representations** We further apply our Gaussian Pruning approach to Scaffold-GS [67], an advanced neural Gaussian-based 3D representation. In this method, we prune Gaussians that contribute minimally to scene reconstruction, reducing redundancy while preserving visual fidelity. Experiments on the MipNeRF360 dataset demonstrate that pruning 80% of neural Gaussians increases the rendering speed from 152 to 178 FPS, as shown in Tab. 5. These results underscore LightGaussian's potential as an effective optimization tool for other Gaussian-based representations.

## 4.3 Ablation Studies

We performed ablation studies on each component of our method to evaluate their individual impacts. Results show that *Gaussian Pruning and Recovery* effectively removes redundant Gaussians, retaining only those crucial for scene representation. *SH Distillation* reduces the Spherical Harmonics degree, simplifying the lighting model with minimal quality loss. Furthermore, *Vector Quantization* efficiently compresses the feature space, creating a more compact representation. Combined, these modules substantially improve the overall efficiency and performance of our framework.

**Significance Criteria** To identify the optimal criteria for measuring each Gaussian's global significance, we evaluated key characteristics: Gaussian-ray interaction count, Gaussian opacity, and functional volume. Tab. 3 illustrates that integrating these three factors — by weighting the hit count with opacity and Gaussian volume — yields the highest rendering quality post zero-shot pruning. A quick Gaussian Co-adaptation (the last row) which optimizes the remaining Gaussians, can effectively recover the rendering accuracy to its pre-pruning level. The visual comparison of rasterized images, before and after pruning, alongside the depiction of pruned Gaussians, is presented in Fig. 5.

**SH Distillation & Vector Quantization.** Directly removing high-degree components in Spherical Harmonics (SHs) causes significant performance degradation compared to the full model (Exp #3 →

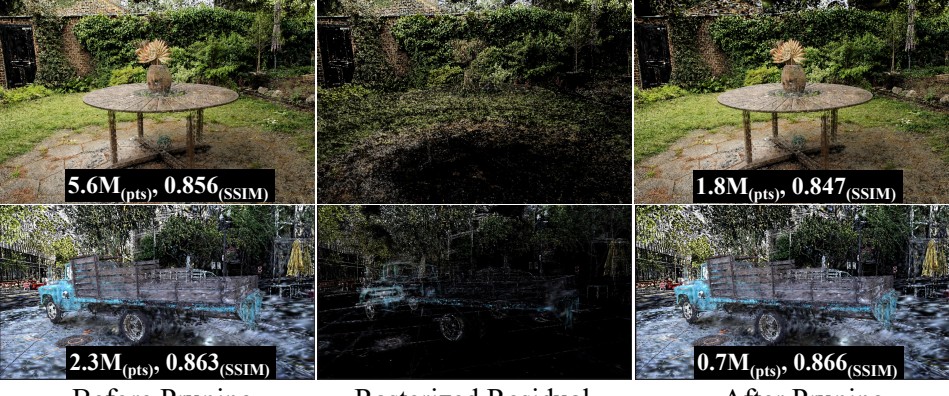

| Before Pruning | Rasterized Residual | After Pruning |

5.6M(pts), 0.856(SSIM)   1.8M(pts), 0.847(SSIM)
2.3M(pts), 0.863(SSIM)   0.7M(pts), 0.866(SSIM)

Figure 5: **Visualization of Pruned Gaussians**. We show the pruned Gaussians (middle) obtained by applying the proposed *Gaussian Prune and Recovery*. The residual is visualized by rasterizing the pruned Gaussians.

4), particularly with the loss of specular reflections across varying viewpoints. However, introducing knowledge distillation from the full model (Exp #4 → 5) allows for size reduction while preserving essential viewing effects. Additionally, incorporating pseudo-views during training (Exp #5 → 6) demonstrates the effectiveness of simulated views in enhancing the learning of specular reflections. While applying VQ to all attributes yields poorer results (see Tab. 4), this impact is mitigated by leveraging Gaussian Global Significance.

| Exp ID | Model setting | Model Size ↓ | FPS ↑ | PSNR ↑ | SSIM ↑ | LPIPS ↓ |
|--------|---------------|--------------|-------|--------|--------|---------|
| [1] | 3D-GS | 353.27 MB | 192 | 31.687 | 0.927 | 0.200 |
| [2] | [1] + 2-degree SH compactness | 140.16 MB | 238 | 30.625 | 0.917 | 0.208 |
| [3] | [2] + proposed distillation | 140.16 MB | 243 | 31.754 | 0.926 | 0.201 |
| [4] | [1] + 1-degree SH compactness | 88.89 MB | 249 | 29.173 | 0.900 | 0.223 |
| [5] | [4] + proposed distillation | 88.89 MB | 244 | 31.440 | 0.923 | 0.205 |

Table 6: Ablation with varying SH compactness: With a Gaussian pruning ratio of 66% and SH reduced to 2 degrees, LightGaussian nearly maintains rendering quality (SSIM drops slightly from 0.927 to 0.926). Reducing SH further to 1 degree lowers SSIM to 0.923.

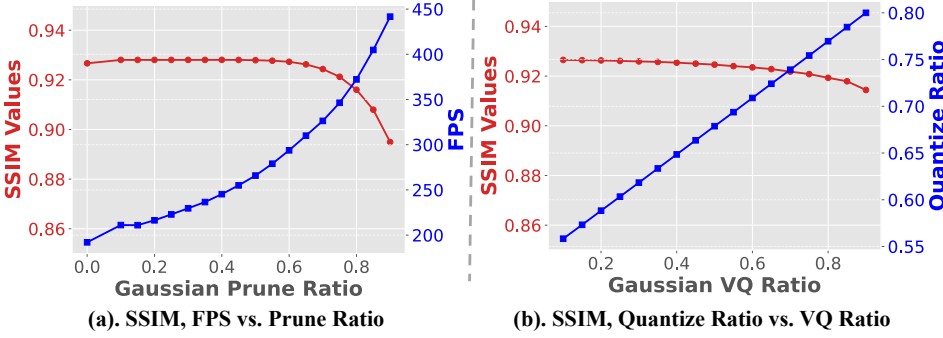

(a). SSIM, FPS vs. Prune Ratio          (b). SSIM, Quantize Ratio vs. VQ Ratio

Figure 6: Rendering performance comparison across different pruning levels and VQ ratios. Note that in subfigure (b), we use the VQ ratio to indicate the proportion of SH parameters quantized to an 8192-codebook, while the (overall) quantization ratio reflects combined compression from VQ and FP32-to-FP16 bitwise quantization.

**Degradation & Speed vs. Compression Ratio**   We investigate the interplay between rendering quality and speed across various model compression ratios. Specifically, we adjust parameters such

as *Gaussian Pruning & Recovery*, *SH Distilling*, and *VQ*. As depicted in Fig 6, we note a marked decline in rendering quality when the pruning ratio reaches 70%, with a more rapid deterioration as the VQ ratio approaches 65%. In Tab 6, we observe that 2-degree SH compactness reduces the SSIM from 0.927 to 0.926, while 1-degree SH compactness further reduces it to 0.923.

## 5   Conclusion, Limitations, and Broad Impact

We present LightGaussian, a novel framework that transforms heavy point-based representations into a compact format for efficient novel view synthesis. Designed for practical use, LightGaussian leverages 3D Gaussians to model large-scale scenes, effectively identifying and pruning the least significant Gaussians generated through densification. It achieves over $15\times$ data reduction, boosts FPS to over 200, and minimally impacts rendering quality. Exploring zero-shot compression across various 3D-GS-based frameworks remains a promising direction for future research.

Broadly, LightGaussian's compact representation has the potential to democratize high-quality 3D content for applications in VR, AR, and autonomous driving by reducing resource demands, enabling more accessible and scalable deployment across industries. While we do not foresee any significant risk from this specific technique, 3D reconstruction may infringe on personal privacy when applied in public spaces or with drone footage, contradicting our ethical intentions.

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

# Appendix

**Algorithm Explanation.** We detail the procedures of LightGaussian in Algorithm 1. The trained 3D-GS [16] features a Gaussian location with a dimension of 3, Spherical Harmonics coefficients with a dimension of 48, and opacity, rotation, and scale, whose dimensions are 1, 4, and 3, respectively.

---

**Algorithm 1** The overall pipeline of LightGaussian

---

**Initialize:** Training view images $\mathcal{I} = \{\boldsymbol{I}_i \in \mathbb{R}^M\}_{i=1}^N$ and their associated camera poses $\mathcal{P} = \{\boldsymbol{\phi}_i \in \mathbb{R}^{3 \times 4}\}_{i=1}^N$.
1: # Pre-Training 3D-GS [16].
2: # $\boldsymbol{G}_i$ with attributes XYZ, SH-3deg, Opacity, Rotation, Scale.
3: $\mathcal{G} = \{\boldsymbol{G}_i \in \mathbb{R}^{(3+48+1+4+3)}\}_{i=1}^N \leftarrow$ 3D-GS$(\mathcal{I}, \mathcal{P})$          ▷ Initial 3D-GS: $\mathcal{G}$
4: #Gaussian Pruning and Recovery: $\mathcal{G} \mapsto \mathcal{G}'$.
5: $\mathcal{GS} \leftarrow$ CALGS$(\mathcal{G})$          ▷ Calculate Global Significance Score $\mathcal{GS}$
6: $\mathcal{G}' \leftarrow$ PRUNE$(\mathcal{G}, \mathcal{GS})$          ▷ Prune Least Significant Ones of $\mathcal{G}$, based on $\mathcal{GS}$
7: $\mathcal{G}' \leftarrow$ RECOVERY$(\mathcal{G}', \mathcal{P})$          ▷ Gaussian Recovery (Finetune $\mathcal{G}'$ on $\mathcal{P}$)
8: #Distilling into Compact SHs: Reduce from 3-degree to 2-degree
9: $\mathcal{G}'' \leftarrow$ REDUCESH$(\mathcal{G}')$          ▷ Reduce the SH degree
10: **while** Few Steps **do**          ▷ SH Distillation
11:     $\hat{\mathcal{P}} =$ SampleView$(\mathcal{P})$          ▷ Synthesize Pseudo Views
12:     $I_t \leftarrow$ TEACHER$(\hat{\mathcal{P}}, \mathcal{G}')$          ▷ Teacher render
13:     $I_s \leftarrow$ STUDENT$(\hat{\mathcal{P}}, \mathcal{G}'')$          ▷ Student render
14:     $\nabla L \leftarrow$ LOSS$(I_s, I_t)$
15:     $\mathcal{G}'' \leftarrow$ ADAM$(\nabla L)$          ▷ Backprop & Step
16: **end while**
17: #Vector Quantization.
18: $\mathcal{G}''' \leftarrow$ VQ$(\mathcal{G}'', \mathcal{GS})$          ▷ VQ based on Significance Score
19: $\mathcal{G}''' \leftarrow$ VQFINETUNE$(\mathcal{G}''')$          ▷ VQ Finetuning
20: #Save Model.
21: Save optimized model $\mathcal{G}'''$ to disk.

---

Specifically, we detail how we calculate each Gaussian's Global Significance Score in Algorithm 2.

---

**Algorithm 2** Global significance calculation for gaussians.

---

1: # $\mathcal{G}$ contains all gaussians with attributes XYZ, SH-3deg, Opacity, Rotation, Scale.
2: # $\mathcal{P}$ Sample cameras from training views.
3: **function** $CalGS(\mathcal{G}, \mathcal{P})$
4:     ScoreList = list()
5:     **for all** Gaussian $G_i$ in $\mathcal{G}$ **do**
6:        score $\leftarrow 0$          ▷ Init Global Significance
7:        count $\leftarrow 0$          ▷ Init Hit Count Value
8:        **for all** Pixel/Ray $r_i$ in RenderFunc$(\mathcal{P})$ **do**
9:           count = GETHITCOUNT$(G_i, r_i)$          ▷ Gaussian Hit count
10:           # Score weighted by opacity/volume
11:           score $\leftarrow$ score + count $\cdot$ opacity$\cdot$ VOLUMEFUNC(scale)          ▷ Eq.3 in draft.
12:        **end for**
13:        ScoreList.append(score)
14:     **end for**
15:     **return** ScoreList
16: **end function**

---

**Post-Finetuning of Vector Quantization.** Applying vector quantization (VQ) based on the significance score facilitates a reduction in the required codebook size compared to the full model. Nonetheless, this method results in a decrease in accuracy as it groups the attributes into several clusters (codes). We observe that the SSIM decreases from 0.923 (#6) to 0.915 (#7), as illustrated in Table 2 of the main draft. This observation motivates us to joint optimize all other attributes and the codes post the VQ. Specifically, by assigning a unique code to each Gaussian, we proceed to implement differentiable finetuning using photometric loss, which is similar to the methodology

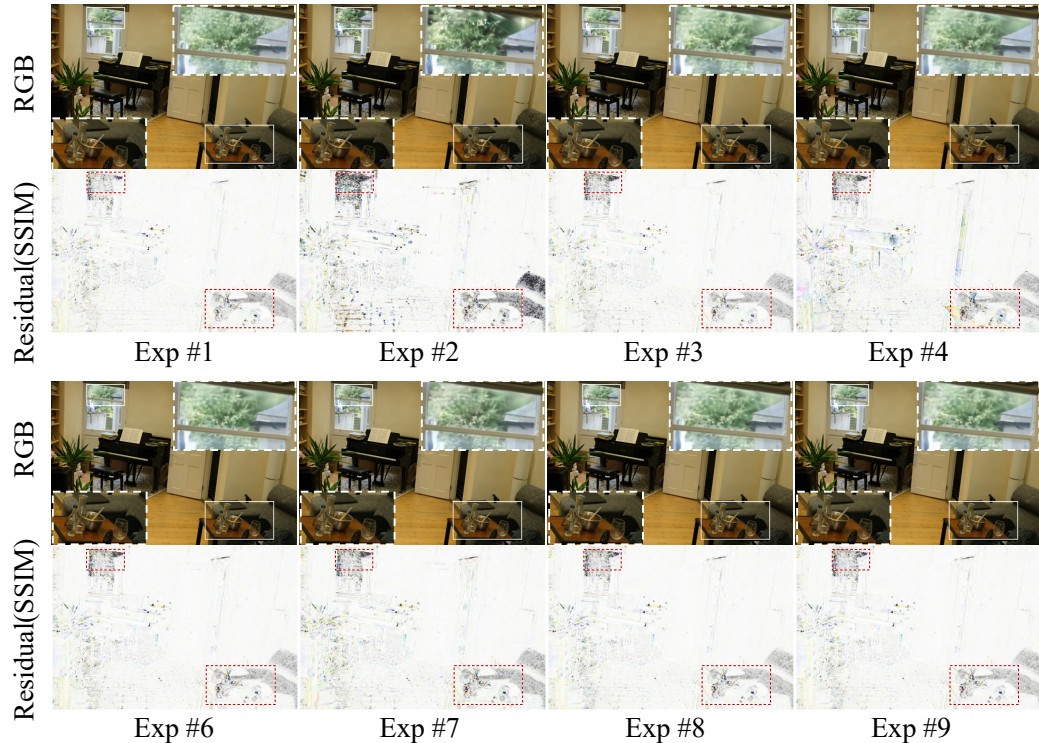

Figure 7: **Visual Comparisons for Ablation Study**. We visualize the rendered RGB images and the residual map between the ground-truth image, aligned with the experiment ID as shown in Tab. The final model (Exp #9) demonstrates close results to 3D-GS (Exp #1), while the Gaussian Co-adaptation, along with SH distillation, almost completely mitigates the information loss. To highlight the difference, we visualize the SSIM between the rendered images and the GT, where a whiter output in the **residual** signifies closer alignment with the GT, denoting better quality, and the area with greater color intensity indicates lower quality.

employed in 3D-GS, and optimize all the attributes in conjunction with VQ. This approach permits precise adjustments to the codebook and the rest attributes, improving its alignment with the training images (#8, SSIM improves from 0.915 to 0.923).

## 6 More Experiment Results

In addition to realistic indoor and outdoor scenes in the Mip-NeRF360 [7] and Tank and Temple datasets [65], we further evaluate our method on the synthetic *Blender* dataset [5], and provide a scene-wise evaluation on all datasets, accompanied by detailed visualizations.

**Results on NeRF-Synthetic 360°(Blender) Dataset.** The synthetic *Blender* dataset [5] includes eight photo-realistic synthetic objects with ground-truth controlled camera poses and rendered viewpoints (100 for training and 200 for testing). Similar to 3D-GS [16], we start training the model using random initialization. Consequently, we calculate the Global Significance of each Gaussian, work to reduce the SH redundancy, and apply the Vector Quantization (codebook size set at 8192) to the learned representation. Overall comparisons with previous methods are listed in Table 7, where we observe our LightGaussian markedly reduces the average storage size from 52.38MB to 7.89MB, while improving the FPS from 310 to 411 with only a slight rendering quality decrease.

**Additional Qualitative Results on Mip-NeRF360.** We provide extra visualizations for 3D-GS [16], LightGaussian (ours), and VQ-DVGO [49], accompanied by the corresponding residual maps from the

Table 7: **Per-scene results on Synthetic-NeRF.**

| Method | Chair | Drums | Ficus | Hotdog | Lego | Materials | Mic | Ship | Avg. |
|---|---|---|---|---|---|---|---|---|---|
| | | | | | Size(MB) | | | | |
| 3D-GS | 94.612 | 64.163 | 35.839 | 39.607 | 64.910 | 29.335 | 34.185 | 56.400 | 52.381 |
| Ours | 13.785 | 9.596 | 5.473 | 5.994 | 9.600 | 4.542 | 5.252 | 8.464 | 7.838 |
| | | | | | PSNR(dB) | | | | |
| 3D-GS | 35.436 | 26.294 | 35.614 | 37.848 | 35.45 | 30.533 | 36.585 | 31.642 | 33.716 |
| Ours | 34.769 | 26.022 | 34.484 | 36.461 | 34.944 | 29.341 | 35.370 | 30.405 | 32.725 |
| | | | | | SSIM | | | | |
| 3D-GS | 0.987 | 0.954 | 0.987 | 0.985 | 0.981 | 0.961 | 0.992 | 0.904 | 0.969 |
| Ours | 0.986 | 0.952 | 0.985 | 0.982 | 0.979 | 0.954 | 0.990 | 0.896 | 0.965 |
| | | | | | LPIPS | | | | |
| 3D-GS | 0.0133 | 0.0405 | 0.0121 | 0.0232 | 0.0195 | 0.0404 | 0.0073 | 0.111 | 0.0334 |
| Ours | 0.0142 | 0.0431 | 0.0137 | 0.0275 | 0.0222 | 0.0461 | 0.0087 | 0.121 | 0.0370 |

Table 8: **Quantitative Comparison (PSNR)** on Mip-NeRF360 and Tank & Temple Scenes. We re-train 3D-GS, Compressed 3D-GS, Compact 3D-GS using their original code and 3D-GS configurations on our platform, to perform a fair comparison.

| Methods | PSNR | | | | | | | | | | |
|---|---|---|---|---|---|---|---|---|---|---|---|
| | bicycle | flowers | garden | stump | room | treehill | counter | kitchen | bonsai | truck(T&T) | train(T&T) |
| Plenoxels | 21.912 | 20.097 | 23.4947 | 20.661 | 22.248 | 27.594 | 23.624 | 23.420 | 24.669 | 23.221 | 18.927 |
| INGP-Big | 22.171 | 20.652 | 25.069 | 23.466 | 22.373 | 29.690 | 26.691 | 29.479 | 30.685 | 23.383 | 20.456 |
| mip-NeRF360 | 24.305 | 21.649 | 26.875 | 26.175 | 22.929 | 31.467 | 29.447 | 31.989 | 33.397 | 24.912 | 19.523 |
| VQ-DVGO | 22.092 | 18.934 | 24.127 | 23.428 | 28.502 | 21.440 | 26.035 | 25.459 | 27.990 | - | - |
| 3D-GS | 25.194 | 21.414 | 27.29 | 26.598 | 31.336 | 22.476 | 28.989 | 31.328 | 31.948 | 25.372 | 21.957 |
| Compressed 3D-GS | 25.040 | 21.126 | 26.817 | 26.357 | 31.072 | 22.298 | 28.653 | 30.755 | 31.161 | 25.194 | 21.882 |
| Compact 3D-GS | 24.795 | 20.974 | 26.703 | 26.260 | 30.645 | 22.570 | 28.553 | 30.332 | 31.84 | 25.041 | 21.544 |
| Ours | 25.196 | 21.538 | 26.961 | 26.77 | 31.399 | 22.685 | 28.478 | 30.869 | 31.414 | 25.399 | 21.838 |

ground truth. As evidenced in Figure 8 and Figure 9, LightGaussian outperforms VQ-DVGO, which utilizes NeRF as a basic representation. Furthermore, LightGaussian achieves a comparable rendering quality to 3D-GS [16], demonstrating the effectiveness of our proposed compact representation.

**Additional Quantitative Results on Mip-NeRF360.** Tables 8, 9, and 10 present the comprehensive error metrics compiled for our evaluation across all real-world scenes (Mip-NeRF360 and Tank and Temple datasets). Our method not only compresses the average model size from 782MB to 45MB, but also consistently demonstrates comparable metrics with 3D-GS on all scenes. LightGaussian additionally shows better rendering quality than Plenoxel, INGP, mip-NeRF360, and VQ-DVGO.

**Implementation Details of VQ-DVGO.** In the implementation of VQ-DVGO [49], we initially obtain a non-compressed grid model following the default training configuration of DVGO [9]. The pruning quantile $\beta_p$ is set to 0.001, the keeping quantile $\beta_k$ is set to 0.9999, and the codebook size is configured to 4096. We save the volume density and the non-VQ voxels in the fp16 format without additional quantization. For the joint finetuning process, we have increased the iteration count to 25,000, surpassing the default setting of 10,000 iterations, to maximize the model's capabilities. All other parameters are aligned with those specified in the original VQ-DVGO paper [49], ensuring a faithful replication of established methodologies.

Table 9: **Quantitative Comparison (SSIM)** on Mip-NeRF360 and Tank & Temple Scenes. We re-train 3D-GS, Compressed 3D-GS, Compact 3D-GS using their original code and 3D-GS configurations on our platform, to perform a fair comparison.

| Methods | SSIM | | | | | | | | | | |
|---|---|---|---|---|---|---|---|---|---|---|---|
| | bicycle | flowers | garden | stump | room | treehill | counter | kitchen | bonsai | truck(T&T) | train(T&T) |
| Plenoxels | 0.496 | 0.431 | 0.606 | 0.523 | 0.509 | 0.842 | 0.759 | 0.648 | 0.814 | 0.774 | 0.663 |
| INGP-Big | 0.512 | 0.486 | 0.701 | 0.594 | 0.542 | 0.871 | 0.817 | 0.858 | 0.906 | 0.800 | 0.689 |
| mip-NeRF360 | 0.685 | 0.584 | 0.809 | 0.631 | 0.745 | 0.910 | 0.892 | 0.917 | 0.938 | 0.857 | 0.660 |
| VQ-DVGO | 0.473 | 0.355 | 0.614 | 0.563 | 0.861 | 0.476 | 0.777 | 0.760 | 0.842 | - | - |
| 3D-GS | 0.764 | 0.601 | 0.862 | 0.770 | 0.917 | 0.632 | 0.906 | 0.925 | 0.939 | 0.878 | 0.811 |
| Compressed 3D-GS | 0.752 | 0.584 | 0.845 | 0.757 | 0.912 | 0.622 | 0.896 | 0.916 | 0.932 | 0.873 | 0.803 |
| Compact 3D-GS | 0.737 | 0.568 | 0.837 | 0.753 | 0.909 | 0.632 | 0.892 | 0.913 | 0.931 | 0.869 | 0.788 |
| Ours | 0.765 | 0.598 | 0.855 | 0.776 | 0.916 | 0.635 | 0.898 | 0.919 | 0.934 | 0.877 | 0.801 |

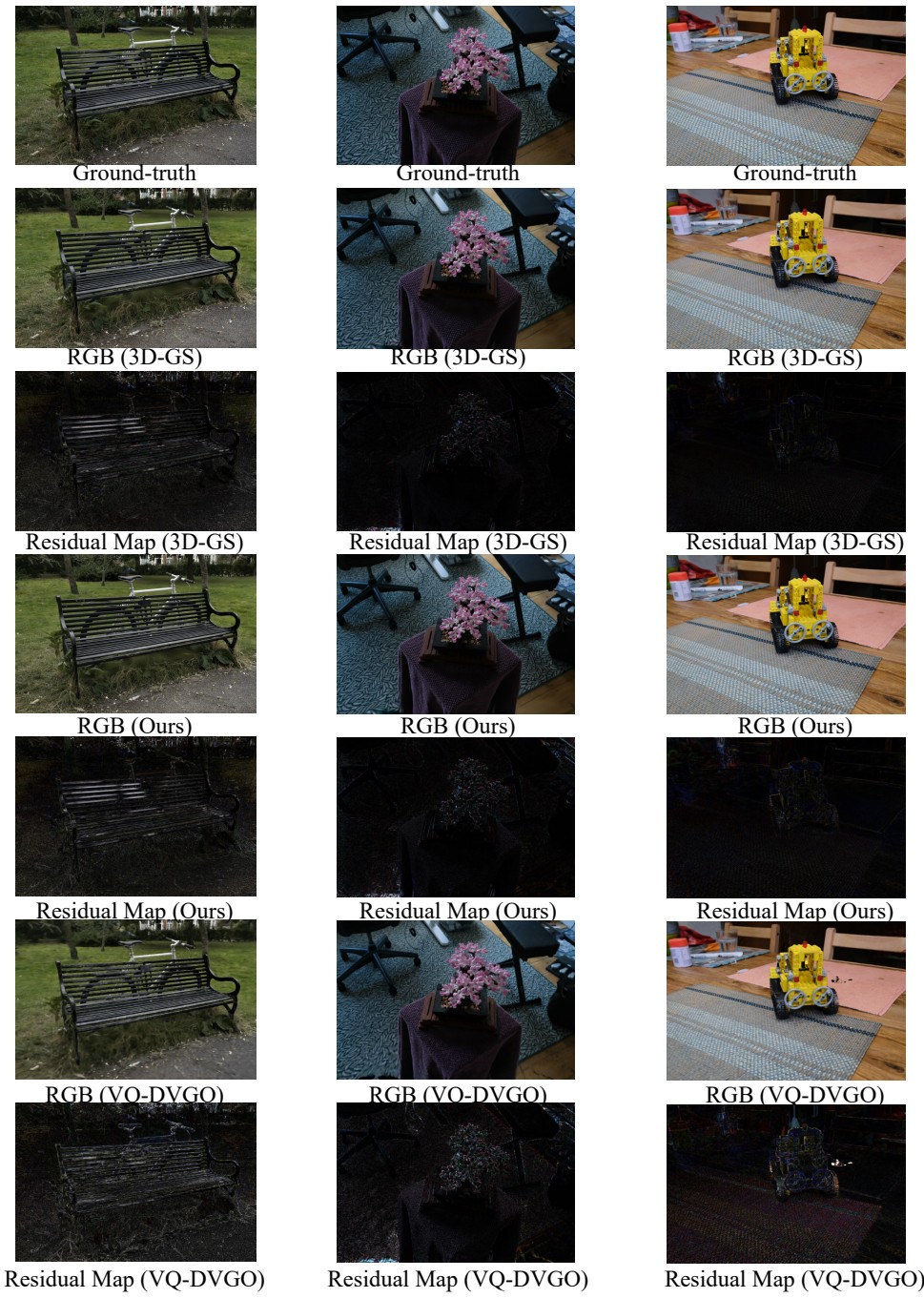

Figure 8: **Additional Visual Comparisons on the Mip-NeRF360 Datasets**. We present the rendering results from 3D-GS [16], LightGaussian, and VQ-DVGO [49]. The residual maps highlight the differences between the rendered images and ground truth (GT) images.

**Overall Analysis.** We report the performance of the investigation on the proposed modules in Tab. 2 and Fig. 7. We verify that our design of *Gaussian Pruning & Recovery* is effective in removing redundant Gaussians (Exp #1→3) with negligible quality degradation while preserving rendering accuracy. This proves that the proposed global significance, based on principle of splatting, accurately represents the critical aspects of each Gaussian. By removing the high-degree SHs and transferring the knowledge to a compact representation (Exp #3 →5), our method successfully demonstrates the benefits from using soft targets and extra data from view augmentation, results in negligible changes

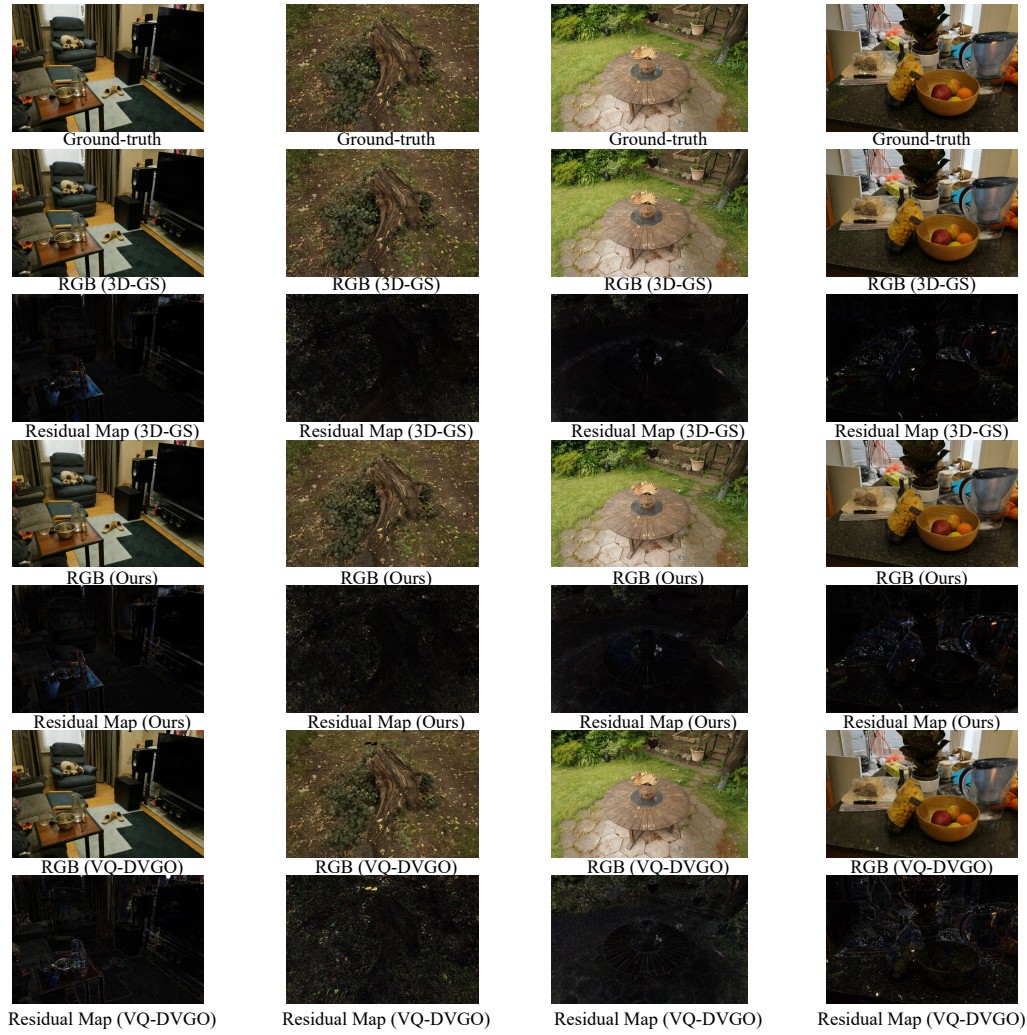

Figure 9: **Additional Visual Comparisons on the Mip-NeRF360 Datasets**. We present the rendering results from 3D-GS [16], LightGaussian, and VQ-DVGO [49]. The residual maps highlight the differences between the rendered images and ground truth (GT) images.

in specular reflection. In practical post-processing, the vector quantization on the least important Gaussians (Exp #7), showcases the advantage of adopting VQ to further reduce model size.

Table 10: **Quantitative Comparison (LPIPS)** on Mip-NeRF360 and Tank & Temple Scenes. We re-train 3D-GS, Compressed 3D-GS, Compact 3D-GS using their original code and 3D-GS configurations on our platform, to perform a fair comparison.

| Methods | LPIPS | | | | | | | | | | |
|---|---|---|---|---|---|---|---|---|---|---|---|
| | bicycle | flowers | garden | stump | room | treehill | counter | kitchen | bonsai | truck(T&T) | train(T&T) |
| Plenoxels | 0.506 | 0.521 | 0.3864 | 0.503 | 0.540 | 0.4186 | 0.441 | 0.447 | 0.398 | 0.335 | 0.422 |
| INGP-Big | 0.446 | 0.441 | 0.257 | 0.421 | 0.450 | 0.261 | 0.306 | 0.195 | 0.205 | 0.249 | 0.360 |
| mip-NeRF360 | 0.305 | 0.346 | 0.171 | 0.265 | 0.347 | 0.213 | 0.207 | 0.128 | 0.179 | 0.159 | 0.354 |
| VQ-DVGO | 0.571 | 0.628 | 0.402 | 0.425 | 0.216 | 0.640 | 0.244 | 0.222 | 0.191 | - | - |
| 3D-GS | 0.211 | 0.339 | 0.108 | 0.216 | 0.221 | 0.327 | 0.201 | 0.127 | 0.206 | 0.147 | 0.209 |
| Compressed 3D-GS | 0.237 | 0.358 | 0.139 | 0.248 | 0.234 | 0.352 | 0.215 | 0.140 | 0.219 | 0.157 | 0.220 |
| Compact 3D-GS | 0.256 | 0.378 | 0.144 | 0.256 | 0.233 | 0.347 | 0.223 | 0.140 | 0.218 | 0.163 | 0.240 |
| Ours | 0.218 | 0.352 | 0.122 | 0.222 | 0.232 | 0.338 | 0.220 | 0.141 | 0.221 | 0.155 | 0.239 |

