# OpenReview forum: "LightGaussian: Unbounded 3D Gaussian Compression with 15x Reduction and 200+ FPS"
_NeurIPS.cc/2024/Conference — NeurIPS 2024 spotlight_

### Official Review · Reviewer_TPCr · 2024-06-14

**Soundness:** 3
**Presentation:** 3
**Contribution:** 2
**Rating:** 6
**Confidence:** 5

**Summary:**

LightGaussian introduces a three-stage technique to efficiently reduce the number of Gaussian primitives. In the first stage, redundant Gaussian primitives are pruned based on global significance, rather than opacity. The second stage involves SH distillation, which utilizes data augmentation from synthesized pseudo views. Finally, the third stage employs Vector Quantization of the SH coefficients. By incorporating these three approaches, LightGaussian achieves a remarkable 15x compression while maintaining competitive rendering quality.

**Strengths:**

1. LightGaussian introduces a novel pruning strategy based on global significance, along with SH Distillation and Vector Quantization techniques, to effectively reduce both primitive redundancy and feature redundancy.
2. LightGaussian functions as a plugin and can be utilized in any GS representation model.
3. The authors conducted a comprehensive ablation study to demonstrate the effectiveness of the proposed method.
4. This method efficiently reduces the number of Gaussian primitives, leading to a significant improvement in rendering speed.
5. The paper is well-written and easy to follow.

**Weaknesses:**

1. **Lack of necessary experimental analysis.** Although the authors compare LightGaussian with Compressed 3D-GS and Compact 3D-GS in Table 1, they fail to provide any analysis in the experimental results section, which is quite peculiar. Moreover, most of the experimental analyses focus on NeRF, which seems unnecessary. Furthermore, in Table 1, it is difficult to distinguish the visual quality performance and compressed capacity between LightGaussian and Compressed 3D-GS, and, strangely, there is no qualitative number provided for the FPS of Compressed 3D-GS.
2. **Lack of novelty.** This method appears to be a post-processing approach of 3D-GS and lacks essential characteristics, unlike Scaffold-GS [1], which proposes a hierarchical 3D Gaussian representation and only stores the information of anchors. It would be beneficial for the authors to compare LightGaussian with Scaffold-GS and its subsequent work, HAC [2], in terms of both experimental results and analysis.
3. **Poor layout formatting.** In Figure 7, the author incorrectly typeset the paper, leading to a compromised reading experience. By the way, the comparison in Figure 6 is not clearly distinguishable, as the eight images appear to be nearly identical.

[1] Scaffold-GS: Structured 3D Gaussians for View-Adaptive Rendering

[2] HAC: Hash-grid Assisted Context for 3D Gaussian Splatting Compression

**Questions:**

1. Please provide a more detailed analysis of compression GS methods, such as Compressed 3D-GS and Compact 3D-GS, in the experimental section. In addition to comparing FPS, it would be valuable to include a comparison of training times as well.
2. It would greatly enhance the paper if the authors compare LightGaussian with Scaffold-GS and HAC in the experimental section for a more comprehensive evaluation.

**Limitations:**

The authors discuss the limitation in Section 5.

---

> ### Author Rebuttal · Authors · 2024-08-07
>
> **[W1]: Analysis over other methods? Why is FPS missing for Compressed 3D-GS?**
> Under a fair experimental setting, we observed that LightGaussian outperforms Compact 3DGS and Compressed 3DGS in 4 out of all 5 metrics on the MipNeRF360 dataset while also running fastest on Tank and Temples datasets. Notably, our method significantly boosts the FPS from 144 (3D-GS) to 237, a 64% improvement. We also surpass Compact 3DGS by 53% and Compressed 3DGS by 55% in FPS.  LightGaussian performs similarly in overall visual quality while preserving thin structures slightly better than other methods, shown in the attached PDF. Please refer to the general response for more details.
>
> **[W2]: LightGaussian and Scaffold-GS**
> We integrated the Gaussian Pruning & Recovery into Scaffold-GS to prune up to 80% redundant neural Gaussians and improve the rendering efficiency. See general response for details.
>
>
> **[W4] Update Figure 6 and Figure 7.**
> We will revise the layout of Figure 7 and update Figure 6 with zoomed-in regions to improve the reading experience for both figures.
>
> **[Q1] FPS and training time comparison with Compressed 3D-GS and Compact 3D-GS**
> Please refer to the general response for training and inference efficiency.
>
>
> **[Q2] Comparison with Scaffold-GS and HAC.**
> Scaffold-GS proposes to initialize a sparse grid of anchor points from SfM points, and tethers a set of neural Gaussians with learnable offsets. This method constrained the distribution of 3D while achieving better reconstruction accuracy. HAC further explores  a structured hash grid to exploit the inherent consistencies among unorganized 3D Gaussians that achieves remarkable compression ratio over 3D-GS.
> We demonstrate the application of LightGaussian upon Scaffold-GS in general response.

---

> > ### Comment · Reviewer_TPCr · 2024-08-08
> >
> > Thanks for the authors' sincere reply and effort.
> > I agree with the performance of LightGaussian, thanks to detailed experiments, and all my concerns have been tackled. I will raise my score to 6.

---

> > > ### Author Response · Authors · 2024-08-13
> > > **Response from the Authors**
> > >
> > > We are pleased that our responses have well addressed your concerns. We sincerely appreciate your willingness to raise your score from 4 to 6 and kindly note that it has not yet been updated in the system.

---

### Official Review · Reviewer_yR9d · 2024-07-05

**Soundness:** 2
**Presentation:** 2
**Contribution:** 2
**Rating:** 6
**Confidence:** 5

**Summary:**

The paper focuses on compressing 3D Gaussian Splatting (3D-GS) models by mainly focusing on reducing the number of Gaussians and compressing the feature size of Gaussians. With three key steps: gaussian pruning and recovery, spherical harmonics distillation, and vector quantization, the paper achieves ~15x reduction in disk usage and ~2x faster rendering speeds with minimal loss in rendering quality. The effectiveness is verified on Mip-NeRF 360, Tank and Temple, and NeRF synthetic datasets.

**Strengths:**

* Compacting the storage cost and rendering speed of 3DGS with minimal loss of rendering quality is an interesting task and has a lot of applications such as rendering on VR devices.
* The paper proposed a heuristic metric to evaluate the importance of each Gaussian, to help identify the less important Gaussian which are then pruned out to reduce the storage cost and accelerate the rendering speed.
* The paper used knowledge distillation after reducing the dimension of the spherical harmonic features to recover the rendering quality.
* The paper did detailed ablation studies on the three key steps and the sub-steps, plus different design choices and hyperparameters, which clearly shows the effectiveness of each components.

**Weaknesses:**

* The proposed vector quantization step lacks novelty, for example both [34] and [36] used vector quantization to compress Gaussian attributes, with minor differences in the method.
* In comparison to the prior 3DGS compression works, the improvement is limited. As shown in Table 1, the improvement on the Mip-NeRF 360 dataset is not clear compared to [34], and the results are worse than [34] on the Tank and Temple dataset in terms of both rendering quality and model size. Besides, why the rendering speeds of [34] are not available?
* The method requires at least two finetuning steps (Gaussian co-adaptation and VQ finetuning) for the compression, beyond the pre-trained 3DGS model, and each finetuning step requires the same iterations as pre-training. Prior work for example [36] can compress 3DGS in an end-to-end manner, and achieve similar performance. How long does the method take to compress one 3DGS model, and in comparison with the 3DGS compression baselines?

In conclusion, the paper is not significantly better than the 3DGS compression baselines. The proposed two novel compression methods such as the Gaussian pruning by a heuristic metric and the knowledge distillation do not show considerable improvement over the baselines in terms of rendering quality, rendering speed, and storage cost.

**Questions:**

Please address the issues I pointed out in the 'weakness' section. Furthermore, I would appreciate clarification on the following questions in the authors' rebuttal.
* Can authors provide the rendering speeds of [34] which are missing in Table 1?
* In Table 2 are the reported FPSs for training (finetuning) or testing? If for testing why the FPS changes after some steps that should only affect the training procedure, for example from step [4] to [5]. Are they just noises?
* In the experiments do you use a universal pruning ratio in the Gaussian pruning step, or is it scene-specific?
* (Minor) In Algorithm 1 line 12 why use G' rather than G as the teacher? As the rendering quality of G' is slightly worse than G. Is it because G' is faster to render?

**Limitations:**

The authors provide a discussion of limitations in the paper.

---

> ### Author Rebuttal · Authors · 2024-08-07
>
> **[W1] Vector quantization step lacks novelty**
> LightGaussian is motivated to design a holistic pipeline that effectively reduces the redundancy in the optimized Gaussians (NxC) for both the primitive count (N) and feature dimension (C). A large number of points will additionally result in slow rendering speed. The heavy attribute size for each primitive is mainly caused by the high-degree Spherical Harmonics used for preserving the view-dependent effect.
> Thus, LightGaussian proposes to identify the Gaussians that contribute the least to the training observations (images) to prune them away. This decreases the model size from 353MB to 116MB while increasing FPS from 192 to 303 (Tab.4). Further redundancy in the feature dimension is addressed through distillation to reduce the degree of Spherical Harmonics with pseudo-view augmentation to preserve the view-dependent effect. The model size is further reduced from 116MB to 77MB (Tab.4).
> Vector Quantization (VQ) is a general technique in image synthesis [1] and neural volumetric video representations [2]. We also utilize VQ with joint fine-tuning to further remove primitive redundancy. Overall, we achieve an average compression rate of over 15x while significantly boosting the FPS from 119 to 209, thanks to the holistic framework. We will moderate the tone of the contribution of Vector Quantization within our overall framework.
>
>
>
> **[W2] Improvement is limited? Improvement is not clear in Table 1. Why are the rendering speeds of [34] not available?**
> Please refer to the general response for the metrics under a fair experimental setting. Our method significantly boosts the FPS from 144 (3D-GS) to 237, a 64% improvement. We also surpass Compact 3DGS by 53% and Compressed 3DGS by 55% in FPS.
>
>
> **[W3] Generalization of LightGaussian? Time Comparison?**
>
> Please refer to the general response.
>
> **[W4] Improvement of Gaussian Pruning and SH Distillation?**
> We kindly argue that all three components contribute to both compactness and rendering efficiency. As shown in our ablation study (Table 2): pruning the least important Gaussians improves the FPS from 192 to 303 (+57%) while reducing the model size from 353MB to 116MB (-67%); SH compactness further reduces the model size to 77MB (-33%). We are hopeful that this new paradigm will disseminate valuable insights for efficient 3D learning.
> We respectfully reiterate another reviewer's comment regarding this point: "The idea of Gaussian Pruning & Recovery and SH distillation is novel, and achieves a better balance between the reconstruction quality, storage usage, and inference speed. " **(Review #bxja)**; “demonstrates great performance on real world scenes.” **(Review #3NHK)**; “LightGaussian introduces a novel pruning strategy based on global significance” **(Review #TPCr)**
>
>
> **[Q1] Rendering speeds of [34] in Table 1?**
> Please refer to the general response.
>
> **[Q2] Why FPS changes from step [4] to [5] in table 2**
> The evaluation is automatically performed after each compression step, rendering 1000 images and reporting the average FPS. Thus, slight discrepancies in each row are caused by the independent evaluation. For rows [7] and [8], where the 3D Gaussians are not further updated, we repeat the numbers from row [6].
>
> **[Q3]: Universal pruning ratio, or scene-specific?**
> It is a universal ratio determined by experiments, as shown in Figure 5.
>
>
> **[Q4] Why use G' rather than G as the teacher?**
> They perform very similarly, with G’ being slightly faster.

---

> > ### Comment · Reviewer_yR9d · 2024-08-08
> >
> > Thanks for the detailed experiments and clarification, all my concerns are addressed, I will raise my score to 6.

---

> > > ### Author Response · Authors · 2024-08-13
> > > **Response from the Authors**
> > >
> > > We are pleased that our responses have well addressed your concerns. We sincerely appreciate your willingness to raise your score from 4 to 6 and kindly note that it has not yet been updated in the system.

---

### Official Review · Reviewer_bxja · 2024-07-07

**Soundness:** 3
**Presentation:** 3
**Contribution:** 4
**Rating:** 8
**Confidence:** 5

**Summary:**

In this paper, the authors delivered a compact 3D Gaussian representation, i.e., LightGaussian for novel view synthesis. There are three technical contributions. Firstly, the authors present Gaussian Pruning and Recovery that measure the significance of each Gaussian to the view quality and then prune 3D Gaussian that has minimal impact on the visual quality. Secondly, the authors propose Spherical Harmonics (SH) distillation that condenses the information from higher-order coefficients to a more compact form. Thirdly, a Vector Quantization step is presented to reduce the redundancy in spatial and lighting representations.

**Strengths:**

i) The technology in this paper is very comprehensive, including the three most mainstream methods of compressing deep learning models, i.e., pruning, distillation, and quantization. Thus, this paper has great engineering significance and can inspire subsequent works of compressing 3DGS from three points of view.

ii) The idea of Gaussian Pruning & Recovery and SH distillation is novel. Computing the score for each 3D Gaussian to represent its significance is fancy and reasonable. The insight of designing Eq. (3) as the score function is interesting. As for the distillation, few people pay attention to the learnable numbers, especially the SH coefficients. Distilling the high-order coefficients with pseudo views also provides a new idea of how to reduce the attributes / learnable parameters of 3D Gaussian.

iii) The proposed LightGaussian achieves a better balance between the reconstruction quality, storage usage, and inference speed. To be specific, in table 1, LightGaussian yields improvements of 0.2 dB in PSNR, 81 fps in inference speed, and 6 MB in storage memory than the state-of-the-art method compact 3DGS on the Mip-NeRF 360 datasets. Meanwhile, LightGaussian achieves faster speed and small model size on the Tank and Template datasets with only a 0.2 dB drop.

iv) The experiments are sufficient. The results in Table 2 clearly demonstrates the effect of each component. The experiments in Table 3, 4, and 5 study the variants of the three compressing techniques. The authors consider very comprehensive situations in these ablation study experiments.

v) The static webpage and video demo in the supplementary are very attractive and Exquisite, which save a lot of time for readers to understand what the authors did in this work.

**Weaknesses:**

i) The story line is not coherent and the motivation may need more explanation. In particular, the three technical contributions, i.e., Gaussian Pruning and Recovery, SH Distillation, and Vector Quantization seem like three independent works. Combining them together without clear motivation make this paper like a technical report. Although I appreciate the work the authors have done, it would be better to improve the writing by establishing a coherent story line and adding clear motivation for the proposed methods.

ii) It is better to add discussion and fair comparison with previous 3DGS compressing works such as compressed 3D-GS [34] and compact 3D-GS [36]. For example, what is the difference between the compress techniques? And what about the performance with the same baseline model?

iii) Some technical details may require more explanation, for example, why designing Eq. (3) like this as the contribution score of each 3DGS? It is interesting to know the process and key insight of the designing process. How you think about this question?

iv) For the main results in table 1, why the Compressed 3D-GS does not have the inference speed? And why LightGaussian is better than SOTA on the Mip-NeRF 360 dataset and performs worse on the Tank and Template datasets? It is better to add analysis and discussion here to explain this performance gap.

**Questions:**

I have a question for the presented SH distillation step

Some existing works just use an view-independent component (also low-order SH coefficients) to represent the color of 3D Gaussian [1]. Did you ever try this method? It is interesting to have a discussion between the SH distillation with this option.

[1] Radiative Gaussian Splatting for Efficient X-ray Novel View Synthesis. In ECCV 2024.

**Limitations:**

Yes, the authors have analyzed the limitations and broader impact of the method in section 4.4 of the main paper.

---

> ### Author Rebuttal · Authors · 2024-08-07
>
> **[W1] Rephrase the motivation.**
> We are motivated by the observation that the efficient point-based representation, 3D-GS, and its many follow-ups perform poorly in model size because they have to store each of the N (usually millions) points. A large number of points typically results in slow rendering speed. Additionally, we found a heavy attribute size for each primitive, especially due to the high-degree Spherical Harmonics used for preserving the view-dependent effect.
> This motivates us to design a holistic pipeline that effectively reduces redundancy in the Gaussians (NxC) for both the primitive count (N) and feature dimension (C). We start by identifying the Gaussians that contribute the least to the training observations (images) and prune them away. Gaussian pruning not only reduces model redundancy but also significantly increases rendering efficiency, as there are fewer Gaussians in the viewing frustum. Further redundancy in the feature dimension is addressed through distillation to reduce the degree of Spherical Harmonics with pseudo-view augmentation to preserve the view-dependent effect. Vector Quantization (VQ) is a general technique that we utilize with joint fine-tuning to further remove primitive redundancy.
> We hope our pipeline is general and can be used as a plug-in tool for many researchers in this field. We will highlight the motivation in the revised introduction.
>
> **[W2] Discussion and fair comparison with Compressed 3DGS and Compact 3DGS.**
> **Compact 3DGS** proposes a learnable mask strategy to reduce the number of Gaussians and utilizes a grid-based neural field to replace the spherical harmonics for a compact representation.
> **Compressed 3DGS** proposes a post-processing method that uses sensitivity-aware vector clustering with quantization-aware training to compress the 3D Gaussians.
> In contrast, **LightGaussian** proposes a visibility-based Gaussian Significance Score to identify the least important Gaussians and introduce a distillation strategy with virtual view augmentation to transfer knowledge from teacher to student with low-degree Spherical Harmonics coefficients.
> Please refer to the general response for a fair comparison.
>
>
> **[W3] Explanation over Eq.3.**
> The motivation behind Eq. 3 is that the densification in 3DGS is based on spatial gradients and can be noisy, as it recovers the full 3D representation from limited training views. This results in an overparameterized scene representation to match the training pixels, which can affect the rendering speed if too many Gaussians are in the viewing frustum. Our insight is to find a general formulation to consider the contribution of each 3D Gaussian to the training views, determined by whether they intersect or not (denoted by $1(G(X), r)$), the opacity, and the volume of a Gaussian and how it impacts the training pixels.
>
>
> **[W4.1] Why FPS for Compressed 3D-GS is missing?**
> It is because we reiterated the metrics from their original paper, which does not provide FPS metrics. We reran the fair comparisons in the tables in General Response.
>
> **[W4.2] Discrepancy of the performance between MipNeRF360 and Tank and Temple?**
> In the main draft (Tab. 1), we use the re-trained 3D-GS, which performs slightly differently from the reported results in 3D-GS. We also reiterate the results from the compared methods in our main draft, as they do not disclose the training details. We conduct fair comparisons to align with the methods, mentioned in general response. It can be observed that LightGaussian has significantly higher FPS while preserving similar visual quantitative metrics compared to the other methods.
>
>
> **[Q1] Applying view-independent RGB similar to X-Gaussian?**
> Thanks for the suggestion! X-Gaussian redesigns a radiative Gaussian point cloud model inspired by the isotropic nature of X-ray imaging, incorporating Differentiable Radiative Rasterization and Angle-pose Cuboid Uniform Initialization for X-ray scanners. However, unlike X-ray imaging, simply utilizing view-independent methods significantly decreases the PSNR from 27.40dB to 24.89dB (MipNeRF360 datasets), which is significantly lower than our method (27.13dB). We will include this discussion in the revision.

---

> > ### Comment · Reviewer_bxja · 2024-08-07
> > **Response to the author rebuttal**
> >
> > Thanks for your reply and effort. All my concerns have been addressed. I keep my original score.

---

> > > ### Author Response · Authors · 2024-08-13
> > > **Response from the Authors**
> > >
> > > Thank you for affirming your positive view of our paper.

---

### Official Review · Reviewer_3NHK · 2024-07-14

**Soundness:** 3
**Presentation:** 3
**Contribution:** 3
**Rating:** 7
**Confidence:** 4

**Summary:**

This manuscript presents a pipeline to drastically reduce the size of pretrained 3D Gaussian splatting models in a way that preserves novel view image fidelity and increases rendering speed. This pipeline consists of three parts: 1) pruning based on an introduced global significance score followed by fine-tuning, 2) reducing the spherical harmonic dimension of the Gaussians and using knowledge distillation to recover the lost specular information, 3) vector quantization of the spherical harmonics of gaussians below a certain global significance score threshold. The global significance score per Gaussian is accumulated over all pixel rays that that intersect that Gaussian and is the sum of the opacity of the Gaussian multiplied by a normalized volume of the Gaussian. Experimental results show a marked decrease in the model size and increase in rendering speeds while image quality is only slightly degraded.

**Strengths:**

S1: This manuscript is well written and easy to follow. Details are provided that should allow for the ability to replicate experiments.

S2: The current size of 3D-GS models makes them unusable in low resource settings — such as VR/AR — and compressing these models will open up many new advancements in downstream applications. The proposed pipeline demonstrates great performance on real world scenes and will be a benefit to researchers working in resource constrained settings.

**Weaknesses:**

**Main weaknesses:**

W1: I’m surprised that contribution per pixel ray is sufficient for an effective the global significance score. It seems like a per Gaussian score would need to account for both the Gaussian’s contribution to the input image pixels AND the location of those pixels in 3D space. A Gaussian observed by multiple cameras should hold more importance as it will be more likely to faithfully reconstruct the 3D scene than a Gaussian only observed by a single camera.

W2: I’m also surprised that the global significance score doesn’t need to account for the exponential drop in contribution to the pixel color due to alpha blending from Gaussians in front of the Gaussian whose score is being computed.

W3: Does the global significance score get updated for all Gaussians along a pixel’s ray or is it updated for only those Gaussians that contribute to the pixel color? I suspect it is the latter, but this should be clearly stated in the manuscript.

W4: It’s surprising to me that recovering the higher degree spherical harmonic representation in the lower spherical harmonic degrees via knowledge distillation and sampled views is needed to maintain high image quality after spherical harmonic reduction.  I’d expect that recovering the color via fine-tuning in the just the training views should work quite well and knowledge distillation and sampled views should be ablated against this.


**Cosmetic issues:**

Eq. 4: Volume should always be positive, so there shouldn’t be a need to take the max of normalized volume and 0.

Line 200: “network” should be “model”, 3D-GS models aren’t networks.

**Questions:**

I’m rating this manuscript as “Accept”. I encourage the authors to engage with the weaknesses I’ve listed and will consider raising my score if they are addressed in the rebuttal.

**Limitations:**

This manuscript has adequately covered the limitations of the proposed pipeline.

---

> ### Author Rebuttal · Authors · 2024-08-07
>
> **[W1 & W2] Does the score need to account for the exponential drop in contribution to the pixel color? Is the score for each Gaussian computed using multiple cameras or a single camera?**
> We thank you for the insightful suggestions. With considering the “exponential drop” in Eq.3, we found the Gaussian Pruning metrics slightly improve on both indoor scene and outdoor unbounded scene. In the table below, we replace the Gaussian opacity $\sigma$ with $\sigma \cdot T$, $\alpha$, and $\alpha \cdot T$, where $T$ is transmittaance and $\alpha$ is parameterized by opacity, scale and covariance. We found the use of opacity perform similar with alpha, while considering the transmittance all slightly improve the pruning effectiveness. We will include a more detailed discussion in the revision. Eq. 3 considers the impact from all training cameras (denoted as M) to formulate the score for each Gaussian.
>
>
> | Scene Bicycle                  | PSNR ↑ | SSIM ↑ | LPIPS ↓ |
> |--------------------------|------------|-----------|------------|
> | opacity  |25.09 |	0.7593|	0.2261|
> | opacity*transmittance | 25.17 |	0.7647 |	0.2196|
> | alpha  |25.08 |	0.7591|	0.2256|
> | alpha*transmittance | 25.15 |	0.7630 |	0.2206|
>
> | Scene Room                  | PSNR ↑ | SSIM ↑ | LPIPS ↓ |
> |--------------------------|------------|-----------|------------|
> | opacity  |31.38 |	0.9144 |0.2364|
> | opacity*transmittance | 31.39 |	0.9158 |0.2320|
> | alpha  |31.35 |0.9135|0.2374|
> | alpha*transmittance |31.38 |	0.9158 |0.2317|
>
>
>
> **[W3] Does the score get updated for all Gaussians along a pixel’s ray or for only those Gaussians that contribute to the pixel color?**
> You are right, its’ the latter and we will clarify more about this in Eq.3 about the criteria 1(G(X_j), r_i) in the Sec.3.2. The intersection will stop when the accumulated opacity reaches 0.9999, same as the way in 3D-GS.
>
>
> **[W4] Why distillation is needed in converting high-order SH to low-order SH?**
> To validate the effectiveness of distillation with augmentation, we ablate the distillation design choices in rows [5] and [6] of Table 2. Additionally, we conducted another experiment on Scene Room (MipNeRF360) that only utilizes photometric loss from the training view to recover the model with low-degree Spherical Harmonics coefficients. The results indicate that simply fine-tuning with photometric error can recover most of the accuracy, while the use of teacher-student distillation with data augmentation can further slightly improve the rendering quality. The table below are based on new configuration mentioned in the general response.
>
>
> | Methods/Metrics                       | PSNR ↑ | SSIM ↑ | LPIPS ↓ |
> |--------------------------------------|--------|--------|---------|
> | Photometric Loss                               | 31.42 | 0.9157 |	0.2338
> | Distillation +  Pseudo-views                       | 31.48 |	0.9167|	0.2313
>
>
>
> In the general machine learning domain, we also noticed literature [1] that studies the use of teacher-student distillation to recover full accuracy for the student. By utilizing "mixup" data augmentation to construct virtual training examples, the distillation process can "generate support points outside the original image manifold," which is beneficial for accuracy recovery.
>
>
>
>
>
> **[W5] Cosmetic issues?**
> We will revise the draft accordingly.
>
>
> **Reference**
> [1]. Knowledge distillation: A good teacher is patient and consistent, CVPR 2022

---

> > ### Comment · Reviewer_3NHK · 2024-08-08
> >
> > W2: I’m glad to see that transmittance helped improve the metrics!
> >
> > W4: If you have not already done so, I’d recommend adding this ablation in the appendix with appropriate links in the main paper.
> >
> > Overall, I’m satisfied and will keep my rating the same.

---

> > > ### Author Response · Authors · 2024-08-13
> > > **Response from the Authors**
> > >
> > > Thank you for affirming your positive view of our paper.

---

### Author Rebuttal · Authors · 2024-08-07

**General Response: Rendering speed of Compressed 3D-GS is missing, and a fair comparison?**
The reason for omitting the FPS of Compressed 3D-GS [34] in the main draft is because we reiterated the metrics from their original paper, which does not provide FPS metrics.
We also respectfully point out that the original 3D-GS utilizes different downsampling ratios for different scenes (see full_eval.py in their GitHub repository) while the compared methods do not clarify their settings. Thus, for a fair comparison, we reran experiments for LightGaussian, Compact 3DGS [36], and Compressed 3DGS [34] on the same platform (NVIDIA A6000) using the default 3D-GS resolution configuration. We observed that our method outperforms Compact 3DGS and Compressed 3DGS in 4 out of all 5 metrics on the MipNeRF360 dataset while also running fastest on Tank and Temples datasets. Notably, our method significantly boosts the FPS from 144 (3D-GS) to 237, a 64% improvement. We also surpass Compact 3DGS by 53% and Compressed 3DGS by 55% in FPS.
We provide visual comparisons among the adopted methods in the attached PDF.


| Data: MipNeRF360                 | FPS ↑  | Size ↓ | PSNR ↑ | SSIM ↑ | LPIPS ↓ |
|--------------------------|-----------|----------|------------|-----------|------------|
| 3D-GS (baseline)  | 144     | 782.11 | 27.40    | 0.8192  | 0.217 |
| Compact 3D-GS   | 154       | 49.40 | 27.01   | 0.7986  | 0.243   |
| Compressed 3D-GS   | 152     | 28.63     | 27.03   | 0.8018  | 0.238     |
| LightGaussian            | 237    | 45.21     | 27.13   | 0.8066  | 0.237       |

| Data: Tank and Temples                  | FPS ↑  | Size ↓ | PSNR ↑ | SSIM ↑ | LPIPS ↓ |
|---------------------------|----------|----------|------------|-----------|--------------|
| 3D-GS (baseline)     | 182    | 431.0 | 23.66     | 0.8445 | 0.178      |
| Compact 3D-GS      | 238   | 39.43   | 23.29    | 0.8285  | 0.202     |
| Compressed 3D-GS| 202   | 17.68   | 23.54    | 0.8380  | 0.189     |
| LightGaussian         | 357    | 25.30  | 23.44     | 0.8318  | 0.202    |


**General Response: Training vs Rendering Speed?**
We report the training and inference efficiency based on the fair experimental setting on MipNeRF360 datasets.

| Method                      | Rendering (FPS) | Train Time (minutes)  |
|-----------------------------|-----------------|-----------------------|
| 3D-GS                       | 144             | 23.8 mins             |
| Compact 3D-GS               | 154             | 33.7 mins             |
| 3D-GS + Compressed 3D-GS    | 152             | 23.8 + 3.8 mins       |
| 3D-GS + LightGaussian       | 237             | 23.8 + 9.0 mins       |


**General Response: Generalization to different methods?**
LightGaussian can function as a plugin and is more general for various point-based representations. Specifically, we also validate LightGaussian on Scaffold-GS [1] to prune redundant neural Gaussians using our Visibility-aware Gaussian Pruning & Recovery. We observe that by pruning 80% of neural Gaussians, we can accelerate the rendering speed of Scaffold-GS from 152 to 173 FPS. Results are averaged on MipNeRF360 datasets. We reran the experiments on our platform for fair comparisons.

| Methods/Metrics                      | FPS | PSNR ↑ | SSIM ↑ | LPIPS ↓ |
|--------------------------------------|-----|--------|--------|---------|
| 3D-GS                                | 144 | 27.40  | 0.8192 | 0.217   |
| Scaffold-GS                          | 152 | 27.96  | 0.8240 | 0.2075  |
| HAC                                  | 167 | 27.76  | 0.8191 | 0.2198  |
| Scaffold-GS + LightGaussian          | 173 | 27.78  | 0.8187 | 0.2197  |



**Reference**
[1] Scaffold-GS: Structured 3D Gaussians for View-Adaptive Rendering, CVPR 2024.

---

### Decision · Program_Chairs · 2024-09-25

**Decision:**

Accept (spotlight)

**Comment:**

Pre-rebuttal, the paper received mixed scores that required a lot of attention from the authors. Recurring issues were related to missing comparisons to existing 3DGS compression methods (in terms of the speed and memory consumption), and to lack of novelty.

Authors did a good job clearing all concerns related to additional comparisons, and they are strongly encouraged to include this additional analysis in the final version. Regarding the novety concerns, the AC agrees that each individual component is not novel on its own, but the significant effectivity of the proposed component marriage is novel.

Post-rebuttal, all reviewers raised their scores above the acceptance threshold which sends the paper well above the acceptance bar.
Authors are encouraged to include the additional performance analysis (requested by all reviewers) in the final version, together with the promised improvements to the exposition (reviewer bxja).